# Mitigation of chytridiomycosis on *Bombina variegata* tadpoles with the antimicrobial metabolites of the bacterium *Xenorhabdus szentirmaii*

János Ujszegi,[1,2] Zsófia Boros,[1,3] Krisztián Harmos,[4] Gábor Magos,[4] Ábris G. Tóth,[1,5] Judit Vörös,[6] Andrea Kásler[1,2,7]

**ABSTRACT**    Survival of amphibian assemblages is threatened by many factors. Among them, chytridiomycosis, the disease caused by the chytrid fungus *Batrachochytrium dendrobatidis* (Bd), has great importance, also threatening populations of the yellow-bellied toad (*Bombina variegata*), which has a scattered, isolated distribution in Hungary. Treatment with secondary metabolites of the entomopathogenic bacterium *Xenorhabdus szentirmaii* is a promising method in the mitigation of chytridiomycosis, as recently demonstrated. To assess whether it may also be effective for lowering infection intensities in *B. variegata* tadpoles, we extracted cell-free culture media (CFCM) from liquid cultures of *X. szentirmaii* and tested their antifungal efficacy at a dilution of 0.1% (v/v), while also measuring possible short-term malign effects on tadpoles experimentally infected with Bd or sham infection. According to our results, CFCM treatment alone did not compromise tadpoles' survival probability, nor reduce the body mass and developmental rate of the individuals. At the same time, the treatment reduced the intensity and prevalence of Bd infection, but only in the case of one population out of the four tested. The antimicrobial metabolites produced by *X. szentirmaii* may therefore be suitable for the safe mitigation of chytridiomycosis in *B. variegata* tadpoles, but further studies are needed, aiming to validate and increase the efficacy of this method.

**IMPORTANCE**  The importance of the study is that it offers a potential new microbial tool in the fight against chytridiomycosis, a major threat to amphibian populations worldwide, including the yellow-bellied toad in Hungary. The research tested a promising method using bacterial metabolites of the bacterium *Xenorhabdus szentirmaii* to combat the fungal chytrid pathogen. This treatment was safe for tadpoles and capable of reducing infection prevalence. While the results are encouraging, there is a need for further research to improve the treatment's effectiveness. This method could help protect vulnerable species like the yellow-bellied toad from the impacts of chytridiomycosis, which is among the most challenging issues in amphibian conservation.

**KEYWORDS**  amphibian disinfection, EPN-EPB symbionts, frog-killing fungus, mitigation method, yellow-bellied toad

**Peer Reviewer** Shalika Silva, Glenville State University, Glenville, West Virginia, USA

Address correspondence to János Ujszegi, ujszegi.janos@gmail.com.

The authors declare no conflict of interest.

See the funding table on p. 10.

Native amphibians are key taxa in many ecosystems serving as prey for predators (1–3), consuming a wide array of invertebrates including pests and vector organisms (4, 5) and playing an important role in ecosystem services (6). Despite their significance, amphibians are going through a biodiversity crisis that started in the last century, becoming the most threatened vertebrate group in the present days (7). The main causes of this decline are climate change, pollution, habitat loss, overexploitation, invasive species and emerging infectious diseases (8). These factors often exert a combined, synergistic negative effect on the populations of amphibian species (9, 10).

The natural defense mechanisms of amphibians are normally effective against a wide range of pathogens and parasites (11), but pathogens introduced by human activities can have devastating effects on naïve amphibian populations. Chytridiomycosis is the most serious infectious disease affecting amphibians (12). The disease is caused by the chytrid fungi *Batrachochytrium dendrobatidis* (Bd) and *Batrachochytrium salamandrivorans* (Bsal), infecting the keratinous epidermal layers of amphibian skin (13). These agents have already contributed to the decline or extinction of several hundred amphibian species and caused mass mortality events on all continents inhabited by amphibians (14). Because Bsal has a much narrower distribution range (15), here we concentrate on the better-investigated and globally distributed Bd. The clinical signs of heavy Bd infection are intensive sloughing or skin shedding, reddening on legs, and ulcerations or skin lesions on ventral surfaces, ultimately loss of righting reflex and body posture. The structural damage to the skin can impair skin breathing and osmoregulation, leading to shifts in electrolyte balance and finally provoking cardiac arrest (16). In general, anuran tadpoles are less susceptible to the disease than in later life stages (14), because keratinous elements are exhibited only in their mouthparts (17), thus they can act as reservoirs in natural habitats (18, 19). However, the presence of Bd sometimes contributes to the loss of keratinized mouthpart structures, therefore leading to reduced feeding abilities and lower survival (20). Reducing Bd infection load during the tadpole stage may mitigate the developmental costs associated with infection and potentially improve survival and fitness during the more vulnerable metamorphic stage (21–23). Several countermeasures against the disease have been proposed so far (24–27), but a widely applicable mitigation method against chytridiomycosis has not been found yet (28, 29), especially for tadpoles.

The addition or supplementation of mutualistic skin bacteria (bioaugmentation) associated with amphibians' skin, that can prevent infections or disease propagation (30, 31) is proposed to be a promising mitigation method against chytridiomycosis (32–34). However, *in vivo* experiments usually reported moderate or no mitigation effect against the disease (29, 35, 36). The target bacteria intended to settle on the new amphibian host can trigger an immune response (34) and may not establish or produce the expected antifungal metabolites due to microbial competition or changing environmental factors experienced on the new host (37–39). Utilizing bacterial metabolites directly against Bd instead of trying to establish live cultures on amphibian hosts has also been tested (40–42). This approach avoids most of the abovementioned problems and can be more safely controlled, and the scope of the search for antifungal metabolites with broad-spectrum inhibition capabilities can be widened to cover microbial sources of non-amphibian origin too (42).

Entomopathogenic bacteria (EPBs) are symbionts of entomopathogenic nematode (EPN) species. These nematodes parasitize insects and release EPBs into the hemocoel of the infected insect hosts. These bacteria start to propagate and synthesize various secondary metabolites that suppress the insect host's immune response and accelerate its death. Furthermore, EPBs produce antimicrobial metabolites that protect the cadaver against microbial food competitors (43, 44). Since EPBs are culturable apart from their EPN hosts, these antimicrobial agents can be further utilized against plant, livestock, and human pathogens (45–49). Moreover, secondary metabolites in cell-free culture medium (CFCM) extracted from liquid cultures of *Xenorhabdus szentirmaii* and *X. budapestensis* EPB species (50) have highly effective growth inhibition capabilities against Bd *in vitro*. In the case of *X. szentirmaii*, CFCM can also be safely applied for significant reduction of Bd infection intensity on juvenile *Bufo bufo* individuals *in vivo* (42). These results highlight the possibility of utilizing EPB metabolites in the mitigation of chytridiomycosis.

In Hungary, at least one Bd strain belonging to the global pandemic lineage (GPL) is present (51) with the highest prevalence in frogs belonging to the *Bombina* and *Pelophylax* genera (52). No mass mortalities have been observed so far, but yearly amphibian population surveys are focusing only on the breeding season. Therefore, mass die-offs during metamorphosis, when amphibians are the most susceptible to

chytridiomycosis (14, 21), may remain hidden since their body size is small, and their carcasses can decompose quickly. In the area of the Bükk National Park (Northern Hungary), a recent study documented several incidents of lethal chytridiomycosis in the case of metamorphosing and juvenile *Bombina variegata* individuals (53). In cooperation with the colleagues of the National Park, as a practical first step of creating a mitigation method against chytridiomycosis in the tadpole stage, we assessed whether *X. szentirmaii* CFCM may be effectively and safely applied on *B. variegata* tadpoles experimentally exposed to Bd. We chose this particular species of bacterium based on its effective Bd growth inhibition *in vitro*, and safe and successful use in amphibians *in vivo* (42).

## MATERIALS AND METHODS

### Culturing of bacteria

We prepared Luria broth agar (LBA) plates flooded with Luria broth (LB) (10 g casein peptone, 5 g yeast extract, 10 g sodium chloride, and 17 g agar dissolved in 1000 mL distilled water) as described previously (54). Indicator plates (Luria Bertani agar; LBTA) were supplemented with bromothymol blue and 2,3,5-triphenyltetrazolium chloride and were used to distinguish antimicrobial metabolite-producing and non-producing variants of *X. szentirmaii* (55). Fresh colonies producing antimicrobial metabolites (indicated by the bluish color of these colonies in contrast with the red color of the non-producing ones) derived from frozen bacterial stocks were used for the experiment as previously described (45–47). Microbiological media were obtained from Biolab Zrt. (Budapest, Hungary).

We cultured *X. szentirmaii* in liquid TGhLY medium (mTGhLY; 8 g tryptone, 2 g gelatine-hydrolysate, 4 g lactose, and 5 g yeast extract in 1000 mL distilled water) with a 7-day-old single colony grown on LBA. We adapted this method from an earlier study (42) to provide optimal growth conditions in the same medium for both the Bd and the EPB with the addition of yeast extract. In all other respects, this medium is equivalent to the TGhL medium used for the culturing of Bd (see below). The EPB culture in this study was started with 5–10 mL of LB inoculated with a single colony of the respective bacterium picked from an LBTA indicator plate and incubated overnight at 28°C in a water bath shaker (Lab-Line Orbital Shaker Water Bath, Marshall Scientific, USA). Each late-log phase inoculum was then added to 200 mL mTGhLY into 400 mL tissue culture flasks to create scale-up cultures.

### Preparation of cell-free culture medium (CFCM)

We incubated scale-up cultures of *Xenorhabdus szentirmaii* for 7 days at 25°C on an orbital shaker platform (Gallencamp, UK). With these preparation conditions, the production of antibiotic metabolites in *Xenorhabdus* cultures reaches its maximum in 5–6 days, containing the same amount of metabolites (45–47). Then, we centrifuged cultures at 6,000 rpm for 20 min at 4°C in 400 mL tubes using a JLA-10.500 type rotor (Avanti centrifuge J-26 XPI, Beckman Coulter, Indianapolis, USA). The supernatant was filtered through a sterile 0.22 µm nylon filter and centrifuged again at the same speed. We considered the resulting supernatant to be a cell-free culture medium (CFCM) of the antibiotic-producing *X. szentirmaii*. To confirm that CFCM was indeed cell-free, we diluted at least two replicates with sterile 2 × LB, incubated them along with the experimental samples, and checked for bacterial growth on LBA plates. We stored CFCM at 4°C in glass bottles until further use.

### Maintaining Bd cultures and experimental exposure

We used the global pandemic lineage (GPL) of Bd. The isolate (Hung_2014) originated from a *B. variegata* collected alive in 2014 by J. Vörös in the Bakony Mountains, Hungary, and isolated by M.C. Fisher and colleagues (Imperial College London, London, UK). We maintained parallel cultures in TGhL medium (mTGhL; 8 g tryptone, 2 g

gelatin-hydrolysate, and 4 g lactose in 1,000 mL distilled water) in 25 cm² cell culture flasks at 4°C and passaged them every 3 months into sterile mTGhL.

One week before performing experimental exposure, we inoculated 100 mL mTGhLY with 2 mL of Bd stock culture in a 175 cm² cell culture flask and incubated it for 7 days at 21°C. We assessed the concentration of intact zoospores (zsp) using a Bürker chamber at ×400 magnification before every inoculation. We inoculated (and re-inoculated after each water change) the tadpoles' rearing water with 1 mL of these cultures, resulting in a final concentration of ~750 zsp/mL in the rearing water. During inoculation, we regularly shook up the Bd culture to distribute the sporangia in the cultures evenly among individuals. We inoculated controls with the same quantity of sterile mTGhLY or RSW according to the treatments. Contaminated water and equipment were disinfected overnight with Virkon S before disposal (24).

## Experimental procedures

In May 2023, we collected 170 *B. variegata* eggs from puddles and wheel tracks at four localities (four distinct populations) in the Mátra mountains, Hungary (Site 1: Haluskási út, 47.8866°N, 19.9938°E; Site 2: Marháti út, 47.8911°N, 20.0418°E; Site 3: Somhegy, 47.8868°N, 20.0076°E; Site 4: Hidasi erdészház, 47.8831°N, 19.9834°E). We transported eggs to the Experimental Station Juliannamajor of the Plant Protection Institute, Center for Agricultural Research located on the outskirts of Budapest (47.5479°N, 18.9349°E). We placed eggs of each of the four populations separately into plastic containers (32 × 22 × 16 cm) holding 0.7 L of reconstituted soft water (RSW: 48 mg $NaHCO_3$, 30 mg $CaSO_4 \times 2\ H_2O$, 61 mg $MgSO_4 \times 7\ H_2O$, 2 mg KCl added to 1 L reverse-osmosis filtered, UV-sterilized tap water (56); at a constant temperature of 19.8 ± 0.4°C and a light dark cycle adjusted weekly to the conditions outside. Nine days after hatching, when all larvae reached development stage 25 (57), we started the experiment with 150 healthy-looking tadpoles. We reared tadpoles individually in opaque plastic boxes (17 × 12 × 9 cm) filled with 1 L RSW and fed them *ad libitum* with boiled and smashed spinach. The temperature in the laboratory was 18.2 ± 0.3°C (mean ± SD) during the experiment. The light:dark cycle was adjusted weekly to outdoor conditions. We assigned tadpoles to five treatments (Table 1), using stratified randomization considering their population of origin: Site 1: N = 45 individuals in total (nine per treatment); Site 2: N = 40 individuals in total (eight per treatment); Site 3: N = 35 individuals in total (seven per treatment); Site 4: N = 30 individuals in total (six per treatment). In the first part of the experiment, we exposed tadpoles to RSW (Treat 1), sterile TGhLY medium (Treat 2 and 3), or liquid culture of Bd at a final concentration of ~750 zsp/mL (Treat 4 and 5) for 22 days (overall 30 individuals per treatment initially, Table 1). We arranged rearing boxes randomly on the laboratory shelves and changed water twice a week during the whole course of the experiment using different dip nets for each treatment to prevent cross-contamination. We monitored survival daily and noted any incidence of death.

In the second part of the experiment, 22 days after the start, we randomly selected five individuals from each treatment group, and humanely euthanized them in a water bath containing 6.6 g/L tricaine-methanesulfonate (MS-222) buffered to neutral pH with the same amount of $Na_2HPO_4$. We conserved euthanized individuals in 96% EtOH and stored samples at 4°C until further analysis as a reference for the initial Bd infection status of the animals before starting the mitigation treatments. We translocated the remaining individuals (25 per treatment) into new rearing boxes with the same

**TABLE 1** Treatment combinations

|         | Culture medium | Bd exposure | CFCM treatment |
|---------|----------------|-------------|----------------|
| Treat 1 | No             | No          | No             |
| Treat 2 | Yes            | No          | No             |
| Treat 3 | Yes            | No          | Yes            |
| Treat 4 | Yes            | Yes         | No             |
| Treat 5 | Yes            | Yes         | Yes            |

parameters containing clear RSW and exposed them to either RSW (Treat 1), sterile mTGhLY (Treat 2 and 4), or CFCM (Treat 3 and 5) at a dilution of 0.1% (v/v) (Table 1). We constantly treated animals for another 22 days re-administering the RSW, medium, or CFCM treatments after each water change. On the last day of the experiment, we gently blotted each individual with a separate paper towel, weighed tadpoles to the nearest mg (OHAUS-PA213 analytical balance, Ohaus Europe Gmb, Nanikon, Switzerland), then euthanized and preserved them as described above.

## Assessment of infection intensity

We cut out, then homogenized the whole mouthparts of the preserved tadpoles, extracted DNA from samples using PrepMan Ultra Sample Preparation Reagent (Thermo Fisher Scientific, Waltham, Massachusetts, USA) according to previous recommendations (58), and stored extracted DNA at −20°C until further analyses. We assessed infection intensity using real-time quantitative polymerase chain reaction (qPCR) following a standard amplification methodology targeting the ITS-1/5.8S rDNA region (58) on a BioRad CFX96 Touch Real-Time PCR System (BioRad Laboratories, Hercules, USA). To avoid PCR inhibition by ingredients of PrepMan, we diluted samples 10-fold with double-distilled water. We ran samples in duplicate, and in case of qualitatively unmatched results (one of the parallels is Bd negative and the other one is Bd positive), we repeated reactions in duplicate. If this again returned an equivocal result, we considered the sample to be Bd positive (59). Genomic equivalent (GE) values were estimated from standard curves based on five dilutions of a standard (1,000, 100, 10, 1, and 0.1 zoospore GE; provided by J. Bosch; Museo Nacional de Ciencias Naturales, Madrid, Spain).

## Statistical analyses

From the analyses, we excluded all the reference individuals preserved for inspecting initial Bd infection status, and a further individual from the infection analyzes which started metamorphosis before the end of the experiment. We assessed treatment effects on survival, the stage of larval development, body mass, Bd infection status (positive or negative for Bd), and infection intensity. For each dependent variable, we ran a model (see model specifications below) with treatment and population of origin (population hereafter) as categorical fixed factors and their interaction. For the analysis of survival, we used Cox's proportional hazards model (R package "coxme") with treatment as the explanatory variable. The structure of the data did not allow for including population as another fixed factor. We entered the number of days until death as the dependent variable, and individuals that survived until the termination of the experiment were treated as censored observations. To analyze variation in the stage of larval development and body mass, we used general linear models (LM; "lm" function of the "nlme" package). In the analysis of body mass, we further included developmental stage as a covariate. We created a binomial variable based on whether an individual was Bd positive or not. Bd prevalence was defined as the proportion of infected individuals. To analyze Bd prevalence, we used generalized linear models ("glm" function of the "nlme" package) with binomial distribution and logit link function. For the investigation of infection intensity, we averaged GE values obtained from qPCR runs for each sample and analyzed resulting estimates corrected for body mass using generalized linear mixed models (GLMM) with negative binomial distribution and a log link function using the "glmmTMB" package (60). All tests were two-tailed, and we checked model fits in the case of all dependent variables by visual inspection of diagnostic plots. We applied a backward stepwise model simplification procedure (61) to avoid potential problems due to the inclusion of non-significant terms (62). In case of significant effect by any grouping explanatory variable, we applied post-hoc tests by calculating pre-planned linear contrasts (63), correcting the significance threshold for multiple testing using the false discovery rate (FDR) method (64). All analyses were conducted in "R" (version 4.0.5).

## RESULTS

Survival was not affected by any of the treatments (Cox model: $z = -1.24$, $P = 0.21$; Fig. 1). The stage of larval development was significantly affected by treatment (LM: $F_{4,94} = 17.19$, $P < 0.001$; Fig. 2A) and by population ($F_{3,94} = 4.27$, $P < 0.001$), but not their interaction ($F_{12,82} = 0.58$, $P = 0.86$). Pairwise comparisons revealed that developmental rate was significantly lower in the group Treat 1 (only RSW) compared with all other groups, but Bd exposure and/or CFCM treatment did not affect it (Table 2). Body mass of the tadpoles at the end of the experiment was not affected by population (LM: $F_{3,92} = 1.41$, $P = 0.25$) or its interaction with treatment (LM: $F_{12,80} = 1.01$, $P = 0.45$), but it was significantly affected by treatment alone ($F_{4,95} = 35.15$, $P < 0.001$; Fig. 2B) and developmental stage (LM: $F_{1,95} = 42.87$, $P < 0.001$). Pairwise comparisons revealed that Treat 1 significantly reduced body mass compared to all other treatment groups. However, Bd exposure or treatment with CFCM did not affect body mass (Table 3, Fig. 2B).

Since all individuals from uninfected treatment groups remained Bd negative, we can exclude the possibility of cross-contamination. In the two Bd-exposed groups, infection prevalence was 70% right before the start of the CFCM treatment, with an average infection intensity of 17.6 (1.0–119.2) GE (median and interquartile range), but this only applies for the reference individuals. CFCM treatment resulted in significantly lower Bd prevalence (GLM: $z = -2.83$, $P = 0.009$; 54% in the treated vs 89% in the non-treated group) and infection intensity ($z = -3.65$, $P < 0.001$; 0.14 [0.0-1.18] GE/gram body weight [GE/gbw; median and interquartile range]; Fig. 3) compared with the Bd-exposed group without CFCM treatment (0.59 [0.13−1.87] GE/gbw [median and interquartile range]; Fig. 3). However, the population also affected this antifungal effect since its interaction with treatment was also significant in the case of infection intensity ($z = 3.58$, $P < 0.001$) and marginally non-significant in the case of Bd prevalence ($z = 1.83$, $P = 0.067$). Omitting this interaction, the effect of population alone on Bd prevalence was significant ($z = 2.29$, $P = 0.022$).

## DISCUSSION

Treatment with cell-free culture media (CFCM) of the *X. szentirmaii* entomopathogenic bacterium (EPB) had no short-term negative effect on the measured life history traits of *B. variegata* tadpoles, similarly to former results on juvenile *B. bufo* individuals exposed to much more concentrated CFCM solutions (42). Since developmental rate and body mass of the tadpoles were significantly lower in the case of individuals reared in pure RSW compared to conspecifics from any other treatments, it seems that microbial medium at the applied dilution exerted a beneficial effect on the measured life history traits (even also containing Bd or bacterial metabolites). This may be due to the extra nutritional

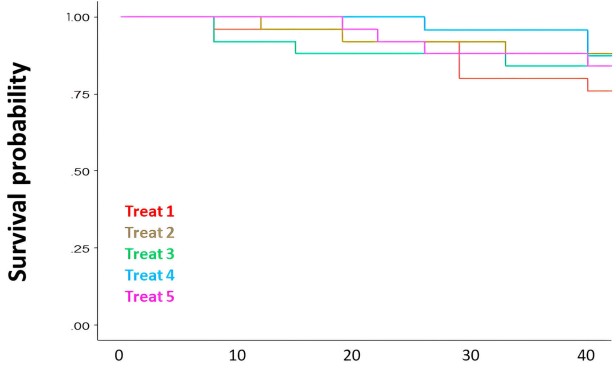

**FIG 1** Survival of *B. variegata* tadpoles over the experiment in the five treatments varying in the presence/absence of culture medium, exposure to Bd, and exposure to bacterially derived metabolites (CFCM). For a summary of treatments, please see Table 1.

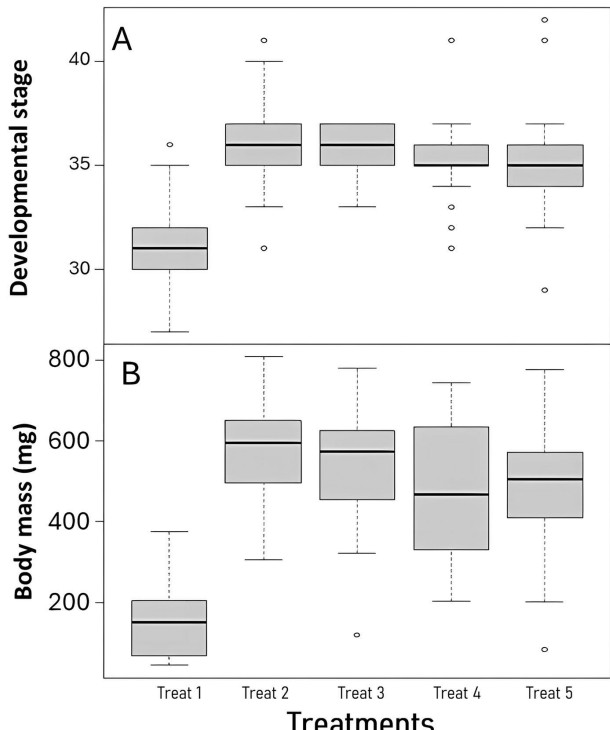

**FIG 2** Responses of *B. variegata* tadpoles to the five treatments in terms of the (A) developmental stage (Gosner) and (B) body mass at the termination of the experiment. In boxplots, horizontal lines and boxes represent medians and interquartile ranges (IQR), respectively, while whiskers extend to IQR ±1.5 × IQR and dots indicate more extreme data points. For the explanation of treatments, please see Table 1.

compounds contained by the medium itself. Tadpoles might have absorbed these nutrients from the water, which accelerated their growth and development. Alternatively, osmolality might have changed due to the addition of medium or the medium exerted a buffering effect, counteracting pH fluctuations due to the $CO_2$ accumulation over time. However, we did not measure these variables; therefore, these alternative explanations remain speculations. Whether CFCM treatment has other long-term negative effects for the treated amphibians, for example, through the alteration of skin microbiome, still needs to be investigated in future studies applying approaches closer to the natural scenarios (e.g. mesocosm experiment).

The prevalence of chytridiomycosis can reach high values in *B. variegata* at several natural habitats across its whole distribution (52, 65–68), likely contributing to local population declines (69). Additionally, mortality events that have been documented (53, 67) further emphasize that intervention may be necessary soon to prevent populations from catastrophic losses. We demonstrated that *X. szentirmaii* CFCM solution applied at a concentration as low as 0.1% v/v is still able to inhibit Bd growth *in vivo* in *B. variegata* tadpoles. This extreme effectiveness is in line with former findings, where dilution close to 0.1% CFCM of the same microbe still showed *in vitro* growth inhibition of Bd (42). However, the beneficial effect of CFCM treatment on infection prevalence and intensity varied between the host's populations of origin, having significant effect on the above variables only in the case of the population from Site 1 (Haluskási-út). While all treated individuals from here completely cleared Bd infection, CFCM treatment had no significant effect on Bd load and infection status on individuals originating from the other three populations. Specimens of *B. variegata* secrete potent AMPs in their skin (70), and intraspecific differences in AMP synthesis can reflect the differences of populations in the sensitivity to chytridiomycosis in other amphibians (71, 72). Therefore, different *B. variegata* populations may differ in their immunity and skin-secreted AMP repertoire, which can affect their resistance to Bd infection and probably differently interact with

**TABLE 2** Pairwise comparisons for the effects of treatments on developmental stage

| Comparison | Hazard ratio ± SE | P |
|---|---|---|
| Treat 1/Treat 2 | −5.285 ± 0.68 | <0.001 |
| Treat 1/Treat 3 | −4.481 ± 0.67 | <0.001 |
| Treat 1/Treat 4 | −4.244 ± 0.67 | <0.001 |
| Treat 1/Treat 5 | −4.148 ± 0.67 | <0.001 |
| Treat 2/Treat 3 | 0.804 ± 0.66 | 0.325 |
| Treat 2/Treat 4 | 1.041 ± 0.66 | 0.194 |
| Treat 2/Treat 5 | 1.137 ± 0.67 | 0.181 |
| Treat 3/Treat 4 | 0.237 ± 0.65 | 0.797 |
| Treat 3/Treat 5 | 0.333 ± 0.65 | 0.763 |
| Treat 4/Treat 5 | 0.096 ± 0.65 | 0.884 |

antimicrobial metabolites in the CFCM. Although these four populations genetically slightly differ based on exome sequencing analyses (73), there may be differences in the degree of hybridization with *B. bombina* and consequently in AMP composition between the individuals originating from these habitats. Our ongoing analyzes about differences in AMP composition of the two *Bombina* species and their hybrids from the study area will presumably provide an answer to this question soon. Also, distinct populations may harbor different skin microbiome compositions (74) variously affecting and interacting with the beneficial effect of the CFCM treatment *in situ*. Nevertheless, we do not think that skin microbiome was strikingly different among these individuals and caused such difference in treatment effect among populations, since we brought these animals into the laboratory as eggs and kept and fed them the same way through the whole experiment, while the composition of skin microbiome is largely habitat-dependent. Improvement of the method's efficacy to reliably achieve more uniform disinfection could help to remove the observed population effect, and future research should aim for a larger sample size to enhance the generalizability of these findings.

One possible option for enhancing efficacy would be the application of a more concentrated CFCM solution, but this can be problematic upon constantly treating tadpoles, exposing them to CFCM through their ambient water. Since CFCM is a culture medium, its ingredients can promote excessive bacterial bloom in the water already at the dilution of 0.5% v/v, which compromises dissolved oxygen level, threatening the tadpoles' health (Ujszegi et al. unpublished). Furthermore, CFCM concentrated more than 15% can be harmful to individuals (42). Taken together, this direction to enhance the effectiveness of CFCM treatment on amphibians is not recommended. An alternative option could be the treatment of tadpoles in separate enclosures out of their rearing water for a limited time on several consecutive occasions to reduce pathogen load or clear infection. In this case, since tadpoles are exposed to CFCM in fresh water on every occasion and only for a limited time, more concentrated CFCM could be used to enhance effectiveness, such as in the case of juvenile *B. bufo* individuals (42). This direction deserves more detailed research and could be a very useful method in the

**TABLE 3** Pairwise comparisons for the effects of treatments on body mass

| Comparison | Hazard ratio ± SE | P |
|---|---|---|
| Treat 1/Treat 2 | −222.35 ± 49.8 | <0.001 |
| Treat 1/Treat 3 | −224.35 ± 46.8 | <0.001 |
| Treat 1/Treat 4 | −169.24 ± 46.3 | 0.001 |
| Treat 1/Treat 5 | −167.00 ± 46.3 | 0.001 |
| Treat 2/Treat 3 | −2.25 ± 38.5 | 0.954 |
| Treat 2/Treat 4 | 53.11 ± 38.6 | 0.216 |
| Treat 2/Treat 5 | 55.35 ± 38.6 | 0.216 |
| Treat 3/Treat 4 | 55.36 ± 37.8 | 0.216 |
| Treat 3/Treat 5 | 57.60 ± 37.8 | 0.216 |
| Treat 4/Treat 5 | 2.24 ± 37.8 | 0.954 |

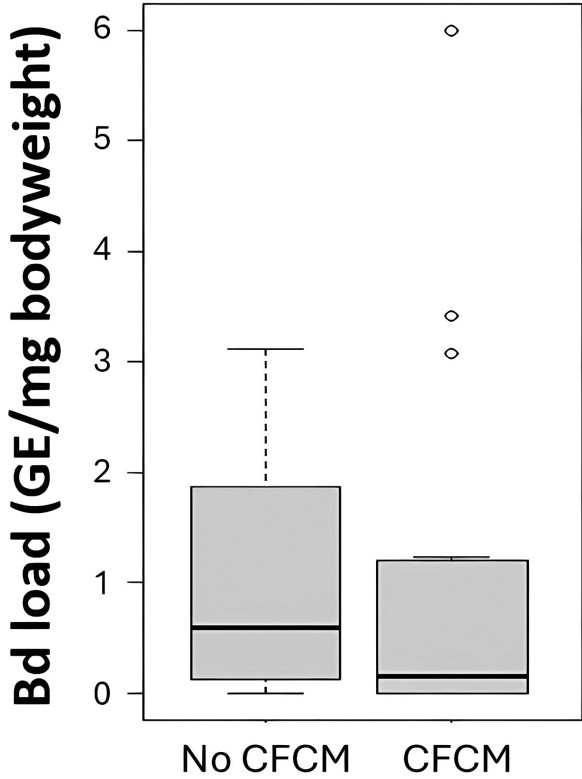

**FIG 3** Tadpoles' infection intensities from the two Bd exposed groups (Treat 4 and Treat 5) in the absence and the presence of 0.1% (v/v) CFCM treatment. Horizontal lines and boxes represent medians and interquartile ranges (IQR), respectively, while whiskers extend to IQR ±1.5 × IQR and dots indicate more extreme data points.

laboratory, or in captive breeding facilities where controlling Bd infection is crucial for preventing disease outbreaks (75), even in the case of *B. variegata* (76). However, for such CFCM treatments, all necessary instruments are better available indoors, and the number of individuals is more manageable than in the field; therefore, *in situ* application of this direction has a very limited potential.

Since a low level of infection usually causes no clinical signs or mortality (77, 78), and amphibian (including *B. bombina*) populations can coexist with Bd (79, 80), a complete clearance of infection may not be essential. Hypothetically, the efficiency achieved by the treatment with 0.1% *X. szentirmaii* CFCM can even be sufficient for the mitigation of chytridiomycosis at *B. variegata* habitats because co-existence with enzootic Bd may lead to immunization (81, 82). However, in the case of *B. variegata,* we do not know any evidence that host-pathogen co-existence may allow for adaptation to the disease via the spread of resistance alleles like in the case of other amphibian species (83–85). Whether the observed effect is sufficient indeed for halting mortality events by Bd in natural populations, and whether the addition of CFCM would be harmful to other aquatic organisms besides amphibians at the habitats (such as pond vegetation and macroinvertebrates) should be assessed and monitored long-term under natural conditions. Given that CFCM has a wide array of antimicrobial activity, and the medium contains many extra nutrients, investigating whether the treatment causes long-term shifts in the aquatic microbial composition and therefore in the flow of energy and nutrients would be also important before routinely applying this method at natural habitats. Whether decreasing Bd intensity in tadpoles has any direct effect on the

severity and consequences of infection or even the infection status in metamorphosed individuals is also a key knowledge gap that requires further research.

A universally applicable mitigation method against chytridiomycosis for most of the host species with the same level of efficacy is not likely to be found (28). Therefore, researchers and nature conservation specialists must focus their effort on species that are endangered or greatly affected by chytridiomycosis. Scientists should find locally adaptable and successful *in situ* mitigation methods designed for the focal species of conservation (86), considering their special needs and characteristics. Going back to *B. variegata*, these frogs mostly live in small water bodies and wheel track puddles. In such environments, the desired CFCM amount (0.1% v/v) could be easily achieved, needing no more than some liters for the treatment of the whole water body. CFCM can still be produced quickly and cheaply in such quantities. Furthermore, due to its extreme thermostability (87), CFCM does not require special handling during transportation and would last for a long time after application. Based on these characteristics and our results, improving this method to be applied for *in situ* mitigation purposes to preserve *B. variegata* populations may be worthwhile in the future.

## ACKNOWLEDGMENTS

We are thankful to A. Hettyey, Z. Mikó, N. Ujhegyi, D. Herczeg, and M. Szederkényi for their assistance and help during the experiment, A. Hettyey for providing the institutional background for the experiment, V. Bókony for help in statistics and M. Z. Németh, and all colleagues from the Department of Plant Pathology for allowing us to use their lab facilities. T. Vellai provided us with the laboratory capacity, and A. Fodor helped with professional guidance at the Department of Genetics, Eötvös Loránd University. Zoospore genomic equivalent standards were kindly offered by J. Bosch, and Bd isolate was sent by M. C. Fisher.

The research was supported by the Lendület Programme of the Hungarian Academy of Sciences (MTA, LP2012-24/2012), the New National Excellence Program and the University Research Fellowship of the Ministry for Innovation and Technology from the source of the National Research, Development, and Innovation Fund (ÚNKP-23-4 and EKÖP-24-4 to U.J., ÚNKP-22-3 and ÚNKP-23-4 to A.K., ÚNKP-23-6 to Á.T.), and by the National Research, Development and Innovation Office of Hungary (NKFIH grant PD-142654 to J.U.).

## AUTHOR AFFILIATIONS

[1]Department of Evolutionary Ecology, HUN-REN Centre for Agricultural Research, Plant Protection Institute, Budapest, Hungary
[2]Department of Systematic Zoology and Ecology, ELTE Eötvös Loránd University, Budapest, Hungary
[3]Department of Genetics, ELTE Eötvös Loránd University, Budapest, Hungary
[4]Bükk National Park Directorate, Eger, Hungary
[5]Department of Zoology, University of Veterinary Medicine Budapest, Budapest, Hungary
[6]HUN-REN Balaton Limnological Research Institute, Tihany, Hungary
[7]Doctoral School of Biology, Institute of Biology, ELTE Eötvös Loránd University, Budapest, Hungary

## AUTHOR ORCIDs

János Ujszegi ⓘ http://orcid.org/0000-0002-6030-0772

## FUNDING

| Funder | Grant(s) | Author(s) |
| --- | --- | --- |
| Ministry of Innovation and Technology | ÚNKP-23-4, EKÖP-24-4 | János Ujszegi |

| Funder | Grant(s) | Author(s) |
|--------|----------|-----------|
| Ministry of Innovation and Technology | ÚNKP-22-3, ÚNKP-23-4 | Andrea Kásler |
| Ministry of Innovation and Technology | ÚNKP-23-6 | Ábris G. Tóth |
| National Research and Innovation Office of Hungary | PD-142654 | János Ujszegi |

## AUTHOR CONTRIBUTIONS

János Ujszegi, Conceptualization, Data curation, Formal analysis, Funding acquisition, Investigation, Methodology, Project administration, Resources, Supervision, Validation, Writing – original draft, Writing – review and editing | Zsófia Boros, Conceptualization, Formal analysis, Investigation, Methodology, Project administration, Validation, Writing – review and editing | Krisztián Harmos, Conceptualization, Investigation, Methodology, Resources, Writing – review and editing | Gábor Magos, Conceptualization, Investigation, Methodology, Resources, Writing – review and editing | Ábris G. Tóth, Data curation, Funding acquisition, Investigation, Methodology, Project administration, Writing – review and editing | Judit Vörös, Conceptualization, Formal analysis, Project administration, Supervision, Validation, Writing – review and editing | Andrea Kásler, Conceptualization, Data curation, Funding acquisition, Investigation, Methodology, Project administration, Resources, Writing – review and editing

## DATA AVAILABILITY

All data sets used in the analyses are available in Figshare at DOI:10.6084/m9.figshare.29313707.

## ETHICS APPROVAL

The Ethical Commission of the Plant Protection Institute approved experimental procedures, and research was carried out according to permits issued by the Government Agency of Pest County (PEI/001/1797-3/2015, PE/EA/00270-6/2023) and the Government Agency of Heves County (HE/TVO/00022-10/2023).

## ADDITIONAL FILES

The following material is available online.

### Open Peer Review

**PEER REVIEW HISTORY (review-history.pdf).** An accounting of the reviewer comments and feedback.

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
