## [Reviewer comments · Microbiology Spectrum]

Microbiology Spectrum

Mitigation of chytridiomycosis on *Bombina variegata* tadpoles with the antimicrobial metabolites of the bacterium *Xenorhabdus szentirmaii*

János Ujszegi, Zsófia Boros, Krisztián Harnos, Gábor Magos, Ábris Tóth, Judit Vörös, and Andrea Kásler

Corresponding Author(s): János Ujszegi, HUN-REN Magyar Kutatasi Halozat

Review Timeline:

Submission Date:	March 19, 2025
Editorial Decision:	April 16, 2025
Revision Received:	June 13, 2025
Accepted:	June 28, 2025

Editor: Renato Kovacs

Reviewer(s): Disclosure of reviewer identity is with reference to reviewer comments included in decision letter(s). The following individuals involved in review of your submission have agreed to reveal their identity: Shalika Silva (Reviewer #2)

Transaction Report:

DOI: <https://doi.org/10.1128/spectrum.00826-25>

Re: Spectrum00826-25 (Mitigation of chytridiomycosis on *Bombina variegata* tadpoles with the antimicrobial metabolites of the bacterium *Xenorhabdus szentirmai*)

Dear Dr. János Ujszegi:

Thank you for the privilege of reviewing your work. Below you will find my comments, instructions from the Spectrum editorial office, and the reviewer comments.

Revision Guidelines

Sincerely,
Renato Kovacs
Editor
Microbiology Spectrum

Reviewer #1 (Comments for the Author):

Line 37, Verb tense disagreement in abstract. "CFM treatment alone did not compromise tadpoles' survival probability, nor reduced the body mass and developmental rate of the individuals." One verb, compromise, is in present tense, while the other verb, reduced, is in past tense. Recommend changing "reduced" to "reduce"

Line 48, change "capable to reduce" to "capable of reducing"

Line 63, change "becoming to be the most" to "becoming the most"

Line 92, change "associated to amphibians' skin" to "associated with amphibians' skin"
Line 96, change "may not establish or not produce" to "may not establish or produce" (the second "not" is not necessary)
Line 104, change parasite (noun) to parasitize (verb)
Line 112, change "In case of" to "In the case of"
Line 120, add a space after the word chytridiomycosis
Line 154, add period after (Gallencamp, UK)
Line 166, add a comma after 2023
Line 194, what is meant by "We changed the boxes" - what was being changed? The water / media? Or both?
Line 198, in "we gently blotted" - blotted with what? Was cross-contamination between individuals or groups prevented?
Line 216, add a space and a ® for the brand name, change "VirkonS" to "Virkon® S"
Line 227, what threshold constituted "unmatched results" - for example, was a 10-fold difference in GE acceptable?
Line 237, what is the difference between "Bd prevalence, and infection intensity"? - Is one positive/negative for Bd and the other is how many zoospores? Suggestion - "Bd infection status" may be better than "Bd prevalence" to indicate positive or negative for Bd.
Line 296, change "Tadpoles might absorbed" to "Tadpoles might have absorbed"
Lines 297-298, suggest changing "other negative effects on the long run" to "other longterm negative effects"
Line 304, remove comma after the word "demonstrated"
Lines 319-320, suggest changing "a more uniform effect" to "a more uniform cohort" or "a more uniform sample population" to avoid using the word "effect" twice in the sentence.
Line 321 and Line 332, the word "effectivity" is confusing. I suggest effectiveness or efficacy instead.
Line 349 - 350, "should be assessed long-term monitored" is not understandable. I suggest "should be assessed and monitored long-term"
Line 356, "is a key knowledge gap and propose further research" doesn't make sense. I suggest "is a key knowledge gap that requires further research."
Line 357, change "The universally applicable mitigation method" to "A universally applicable mitigation method"
Lines 366-367, "due to its extreme thermostability, CFCM does not require special 366 handling during transportation and would last for a long time after application." Is there a citation for this where CFCM has been studied over time at multiple temperatures?
Line 457, italicize *Batrachochytrium salamandrivorans*
Line 461 and Line 679, Voyles et al. 2009 and Voyles et al, 2018 references have "(80-)" after the journal title - it is not clear what this refers to. It does not appear in other citations for this paper. It is not volume, or issue, or page numbers.
Line 491, italicize *In vitro*
Line 538, italicize *Xenorhabdus*
Line 561, remove the space before the comma in "species , *Xenorhabdus szentirmaii*"
Line 561, italicize *Xenorhabdus szentirmaii*
Line 562, remove the space before the period in "*X. budapestensis*"
Line 562, italicize *X. budapestensis*
Line 580, italicize *Batrachochytrium dendrobatidis*
Line 605, remove extra space before semicolon
Line 629, italicize *Bombina variegata*
Line 709, change "Treat5" to "Treat 5" by adding a space.

Remaining points to address:

- 1) In which population(s) was the CFCM effective vs. not? Which sites?
- 2) Can hybrids of the two *Bombina* species be differentiated from *B. variegata*? Or could hybrids have been present within your 150 individuals? If present, is there a way to reliably estimate how many hybrids could have been in your treatment groups?
- 3) Another point to mention in the discussion... In addition to skin AMP differences, the skin microbiomes are also presumably different amongst the four populations.

Reviewer #2 (Comments for the Author):

Comments are attached as a separate file.

Reviewer Comments for Manuscript Spectrum00826-25 in Microbiology Spectrum

- Line 37, Verb tense disagreement in abstract. “CFCM treatment alone did not compromise tadpoles’ survival probability, nor reduced the body mass and developmental rate of the individuals.” One verb, compromise, is in present tense, while the other verb, reduced, is in past tense. Recommend changing “reduced” to “reduce”
- Line 48, change “capable to reduce” to “capable of reducing”
- Line 63, change “becoming to be the most” to “becoming the most”
- Line 92, change “associated to amphibians’ skin” to “associated with amphibians’ skin”
- Line 96, change “may not establish or not produce” to “may not establish or produce” (the second “not” is not necessary)
- Line 104, change parasite (noun) to parasitize (verb)
- Line 112, change “In case of” to “In the case of”
- Line 120, add a space after the word chytridiomycosis
- Line 154, add period after (Gallencamp, UK)
- Line 166, add a comma after 2023
- Line 194, what is meant by “We changed the boxes” - what was being changed? The water / media? Or both?
- Line 198, in “we gently blotted” - blotted with what? Was cross-contamination between individuals or groups prevented?
- Line 216, add a space and a ® for the brand name, change “VirkonS” to “Virkon® S”
- Line 227, what threshold constituted “unmatched results” - for example, was a 10-fold difference in GE acceptable?
- Line 237, what is the difference between “Bd prevalence, and infection intensity”? - Is one positive/negative for Bd and the other is how many zoospores? Suggestion - “Bd infection status” may be better than “Bd prevalence” to indicate positive or negative for Bd.
- Line 296, change “Tadpoles might absorbed” to “Tadpoles might have absorbed”
- Lines 297-298, suggest changing “other negative effects on the long run” to “other longterm negative effects”
- Line 304, remove comma after the word “demonstrated”

- Lines 319-320, suggest changing “a more uniform effect” to “a more uniform cohort” or “a more uniform sample population” to avoid using the word “effect” twice in the sentence.
- Line 321 and Line 332, the word “effectivity” is confusing. I suggest effectiveness or efficacy instead.
- Line 349 - 350, “should be assessed long-term monitored” is not understandable. I suggest “should be assessed and monitored long-term”
- Line 356, “is a key knowledge gap and propose further research” doesn’t make sense. I suggest “is a key knowledge gap that requires further research.”
- Line 357, change “The universally applicable mitigation method” to “A universally applicable mitigation method”
- Lines 366-367, “due to its extreme thermostability, CFCM does not require special 366 handling during transportation and would last for a long time after application.” Is there a citation for this where CFCM has been studied over time at multiple temperatures?
- Line 457, italicize *Batrachochytrium salamandrivorans*
- Line 461 and Line 679, Voyles et al. 2009 and Voyles et al, 2018 references have “(80-)” after the journal title - it is not clear what this refers to. It does not appear in other citations for this paper. It is not volume, or issue, or page numbers.
- Line 491, italicize *In vitro*
- Line 538, italicize *Xenorhabdus*
- Line 561, remove the space before the comma in “species , *Xenorhabdus szentirmaii*”
- Line 561, italicize *Xenorhabdus szentirmaii*
- Line 562, remove the space before the period in “X . *budapestensis*”
- Line 562, italicize *X. budapestensis*
- Line 580, italicize *Batrachochytrium dendrobatidis*
- Line 605, remove extra space before semicolon
- Line 629, italicize *Bombina variegata*
- Line 709, change “Treat5” to “Treat 5” by adding a space.

Remaining points to address:

- In which population(s) was the CFCM effective vs. not? Which sites?

- Can hybrids of the two *Bombina* species be differentiated from *B. variegata*? Or could hybrids have been present within your 150 individuals? If present, is there a way to reliably estimate how many hybrids could have been in your treatment groups?
- Another point to mention in the discussion...In addition to skin AMP differences, the skin microbiomes are also presumably different amongst the four populations.

Title: Mitigation of chytridiomycosis on *Bombina variegata* tadpoles with the antimicrobial metabolites of the bacterium *Xenorhabdus szentirmaii*

In this study, Ujszegi et al. show the effects of metabolites from *Xenorhabdus szentirmaii* bacteria on treating *Bombina variegata* tadpoles with *Batrachochytrium dendrobatidis* infection. These authors have published a work that showed the effects of *X. szentirmaii* metabolites on juvenile toads of *Bufo bufo* in 2023 and this work is an extension of that work. They have tested whether the cell-free culture media (CFCM) of bacteria affects the survival, development, and body mass of the tadpoles and checked whether the CFCM can reduce the Bd prevalence and infection intensities of tadpoles. Authors have shown that CFCM can reduce the intensity and the Bd prevalence of the infected tadpoles without compromising the survival, development, or body mass.

Bacterial secondary metabolites have been used in treating different infections caused by bacteria and fungi however using the metabolites from *X. szentirmaii* in treating chytridiomycosis is a novel and interesting approach for mitigating chytridiomycosis. There are several concerns.

Minor concerns:

- Organizing the methods section to have a flow to make it easy for the readers to understand the approach of the work.
- Line 135 – please spell out the LBTA as this is the first time you say that.
- Lines 135-137 – state indicator media used for identification of antimicrobial peptide producers and non-producers of the bacteria *X. szentirmaii* and labeled as phase 1 and II. It's not clear what are these labeling and did not notice these being used again in the manuscript. I suggest stating how indicator media are used to recognize antimicrobial peptide-producing bacteria.
- Line 148 – incubated *X. szentirmaii* at 28C then in line 153 – have incubated at 25C. Why have you used two different temperatures for growing the bacteria and extracting the CFCM? Please clarify and indicate if you have seen higher CFCM production at lower temperatures.
- Line 155 – I suggest stating that “antimicrobial metabolite production reached its maximum” instead of saying “stationary phase”
- Suggest selecting headings for sections appropriately. In line 131 - started methods section is “materials and methods” and again named a subsection as “experimental procedures” (in line 165). This confuses the readers.
- There is confusion about their labeling of TGhL media. Line 206 – states TGhL and om brackets say mTGhL for the regular TGhL recipe. Authors have to carefully check their labeling of media.

- I suggest authors carefully check the methods section again consolidate some information and reorganize the sections to improve the clarity of the approach.

Major concerns:

- The major concern I have is the sample size that the authors have used for the experiment. According to the authors, they have used different numbers of tadpoles in each treatment for each site. In part 1 of the experiment, they kept 9/treatment for site 1; 8/treatment for site 2; 7/treatment for site 3; and 6/treatment for site 4. Then they continued part 2 of the experiment after 22 days of part 1 after removing five individuals from each treatment group (lines 189-191) and euthanizing them for detecting the Bd infection status of tadpoles. The number of individuals for each treatment for a site is low and if they have removed 5 tadpoles from each treatment group that brings down the number of tadpoles for testing CFCM effects really low. If they have used only 1 individual from each treatment group for detecting Bd infection status that number is still low for detecting Bd infection with higher statistical significance. This drops the rigor of their work. However, the experimental design is good and appropriate.
- Lines 212-214 – describe inoculating tadpoles with 750 zoospores/ml. However, the culture they used for this appeared to be a mixed culture of Bd as they have used 7d old Bd liquid culture. Authors need to be clear about how zoospores were separated (whether they separated the zoospores from the mixture before counting). The 7d old Bd mixed culture grown in culture flasks has more sporangia than spores and that may result in underestimation of the zoospore counts and different infection status.
- Authors do not describe using an internal standard for the qPCR experiment and need one for the accuracy of the experiment which would help in accounting for the sample preparation errors. Having an internal standard is very important as they compare the treatments and sites.
- Line 271-272 – The authors claim CFCM did not affect the body mass of Bd-exposed or CFCM-treated tadpoles. However, Bd-exposed tadpoles have shown slightly reduced body mass than non-Bd-exposed tadpoles. In treatment 2,3, and the CFCM treated group, treatment 5 has slightly higher body mass than 4. So CFCM has shown a slight effect on body mass even though statistically significant. The low number of individuals used might have caused this. I suggest the authors describe the observed results and then state the statistical significance.
- Even though the authors describe the Bd prevalence it is not clear how did they measure the Bd prevalence in the methods section. The authors need to clarify that in the methods section. The data presented in the paper showing the effects of

population on the Bd prevalence and infection intensity would be good. Suggest adding those figures.

- The observed significantly lower growth of the tadpoles housed with reconstituted soft water has not been addressed well in the discussion. Authors have tried to support this difference stating that it might be due to the nutrient media in the treatments. Even though microbial media can have some effects providing some nutrients to tadpoles, it is difficult to accept that this is the major reason for the very prominent growth reduction observed in tadpoles reared in RSW. The method used to make RSW might have affected the growth of the tadpoles and authors need to address this.
- The discussion section needs the author's attention and suggests revising it. The conclusion of the study is not well articulated and not clear as authors make different arguments that contradict each other.
- The authors have stated in the abstract that the reduction of the intensity and Bd prevalence by CFCM was noticed only in the case of one population out of the four populations that they used. However, they have not shown this data or discussed this in the discussion. They have not stated which population was affected by the CFCM. It will be interesting to understand this.
- I suggest authors test the effects using just one population as that would remove the population variability.

Dear Dr. Renato Kovacs,

Thank you for handling our manuscript "Mitigation of chytridiomycosis on *Bombina variegata* tadpoles with the antimicrobial metabolites of the bacterium *Xenorhabdus szentirmaii*" (Spectrum00826-25) submitted to Microbiology Spectrum by János Ujszegi, Zsófia Boros, Krisztián Harmos, Gábor Magos, Ábris Tóth, Judit Vörös and Andrea Kásler. On 16 April 2025 you invited us to revise our manuscript and now we have done that.

Our revision addresses all comments raised by the Reviewers. Our responses are highlighted in italicized text. References cited in the responses are cited at the end of the rebuttal letter. Line numbering corresponds to the original, new version of the document (where changes are not tracked).

We would like to thank the Reviewers for their thoughtful and constructive suggestions; the manuscript is certainly improved as a result of their efforts. We also made some minor corrections on the text. All changes are highlighted in a separate document.

I confirm that none of the material in this manuscript is being considered for publication elsewhere.

Thank you for your time and interest.

Sincerely,

János Ujszegi

Responses to comments

Reviewer #1

We thank the Reviewer for this positive evaluation.

Line 37, Verb tense disagreement in abstract. "CFM treatment alone did not compromise tadpoles' survival probability, nor reduced the body mass and developmental rate of the individuals." One verb, compromise, is in present tense, while the other verb, reduced, is in past tense. Recommend changing "reduced" to "reduce"

We thank for this remark. We changed in the text.

Line 48, change "capable to reduce" to "capable of reducing"

We also changed this accordingly.

Line 63, change "becoming to be the most" to "becoming the most"

We have changed this accordingly.

Line 92, change "associated to amphibians' skin" to "associated with amphibians' skin"

We have changed this accordingly.

Line 96, change "may not establish or not produce" to "may not establish or produce" (the second "not" is not necessary)

Second "not" has been deleted.

Line 104, change parasite (noun) to parasitize (verb)

We executed this change.

Line 112, change "In case of" to "In the case of"

Changed too.

Line 120, add a space after the word chytridiomycosis
Space has been added.

Line 154, add period after (Gallencamp, UK)
Added.

Line 166, add a comma after 2023
Added.

Line 194, what is meant by "We changed the boxes" - what was being changed? The water / media?
Or both?
We translocated the remaining individuals into new rearing boxes. Of course, this means that the water has been also changed to clear RSW before adding CFCM or media according to treatments. We have reworded this sentence to be clearer (lines 218-221).

Line 198, in "we gently blotted" - blotted with what? Was cross-contamination between individuals or groups prevented?
We used a separate paper towel for each individual, this prevented cross contamination between them. We added this information to the MS (lines 223-224).

Line 216, add a space and a ® for the brand name, change "VirkonS" to "Virkon® S"
We changed this.

Line 227, what threshold constituted "unmatched results" - for example, was a 10-fold difference in GE acceptable?
We did not experience such a high difference among samples. Here we meant those cases in general, when one of the parallels is Bd negative and the other one is Bd positive. These cases typically occur at low GE values, when the Bd genome is very diluted in the sample. We made this clear in the description now (lines 236-239).

Line 237, what is the difference between "Bd prevalence, and infection intensity"? - Is one positive/negative for Bd and the other is how many zoospores? Suggestion - "Bd infection status" may be better than "Bd prevalence" to indicate positive or negative for Bd.
Yes, we meant the same. Therefore, we have changed this according to the Reviewer's suggestion.

Line 296, change "Tadpoles might absorbed" to "Tadpoles might have absorbed"
We modified this in the text.

Lines 297-298, suggest changing "other negative effects on the long run" to "other longterm negative effects"
This has been changed.

Line 304, remove comma after the word "demonstrated"
Removed.

Lines 319-320, suggest changing "a more uniform effect" to "a more uniform cohort" or "a more uniform sample population" to avoid using the word "effect" twice in the sentence.
We reworded this to avoid word repetition.

Line 321 and Line 332, the word "effectivity" is confusing. I suggest effectiveness or efficacy instead.
We changed effectivity" to "efficacy" (line 348) and "effectiveness" (line 359).

Line 349 - 350, "should be assessed long-term monitored" is not understandable. I suggest "should be assessed and monitored long-term"
We modified this part accordingly.

Line 356, "is a key knowledge gap and propose further research" doesn't make sense. I suggest "is a key knowledge gap that requires further research."
We modified this part according to the suggestion (line 383).

Line 357, change "The universally applicable mitigation method" to "A universally applicable mitigation method"
Changed.

Lines 366-367, "due to its extreme thermostability, CFCM does not require special 366 handling during transportation and would last for a long time after application." Is there a citation for this where CFCM has been studied over time at multiple temperatures?
Yes, we added a reference proving that even autoclaving leave the antifungal activity of X. szentirmaii CFCM unaltered (Cimen et al., 2021)(line 393).

Line 457, italicize *Batrachochytrium salamandrivorans*

Line 461 and Line 679, Voyles et al. 2009 and Voyles et al, 2018 references have "(80-)" after the journal title - it is not clear what this refers to. It does not appear in other citations for this paper. It is not volume, or issue, or page numbers.

Line 491, italicize *In vitro*

Line 538, italicize *Xenorhabdus*

Line 561, remove the space before the comma in "species , *Xenorhabdus szentirmaii*"

Line 561, italicize *Xenorhabdus szentirmaii*

Line 562, remove the space before the period in "*X. budapestensis*"

Line 562, italicize *X. budapestensis*

Line 580, italicize *Batrachochytrium dendrobatidis*

Line 605, remove extra space before semicolon

Line 629, italicize *Bombina variegata*

We thank the Reviewer for checking the references such thoroughly, we fixed all these issues.

Line 709, change "Treat5" to "Treat 5" by adding a space.
Space has been added.

Remaining points to address:

1) In which population(s) was the CFCM effective vs. not? Which sites?

*CFCM treatment was effective in the case of the population from Site 1 (Haluskási-út). All treated individuals from here completely cleared *Bd* infection. However, CFCM had no significant effect on *Bd* infection intensity and infection status on individuals originating from the other three populations. We added this information to the text (lines 327-329).*

2) Can hybrids of the two *Bombina* species be differentiated from *B. variegata*? Or could hybrids have been present within your 150 individuals? If present, is there a way to reliably estimate how many hybrids could have been in your treatment groups?

*Adult hybrids can be separated from pure *B. variegata* individuals based on morphology in most cases, but it is very challenging, and almost impossible, when they are backcrossed with *B. variegata* over several generations. Hybrid tadpoles, on the other hand, cannot be determined at all based on morphological features. Our study site is the highest region occupied by *B. variegata* in the Mátra mountains, but interestingly a low proportion of *B. bombina* alleles (about 10%) can still be found (Zacho, 2023). Therefore, these *B. variegata* populations are not 100 % pure, thus we cannot surely conclude that all the 150 tested individuals were pure *B. variegata* specimens. It is more likely, that ca. 10 % of them were hybrids for some degree, based on information from other populations of this region (Zacho, 2023). Despite the possibility of this introgression, we think it was the best option to*

collect individuals from here since the most striking examples of mortalities due to chytridiomycosis in Hungary occurred in these populations. The importance of testing the exact populations, that are in need of working out a mitigation method is further supported by the differences in CFCM treatment effectiveness among populations we experienced in our experiment.

3) Another point to mention in the discussion... In addition to skin AMP differences, the skin microbiomes are also presumably different amongst the four populations.

The skin microbiome may be important and could cause different outcomes indeed in situ, but we brought these animals into the laboratory as eggs, and kept and fed them all the same way. Therefore it is very unlikely, that skin microbiome was strikingly different among these individuals, and caused such a difference among populations, since it is largely habitat-dependent. We incorporated this explanation into the Discussion (lines 339-345).

Reviewer #2

In this study, Ujszegi et al. show the effects of metabolites from *Xenorhabdus szentirmaii* bacteria on treating *Bombina variegata* tadpoles with *Batrachochytrium dendrobatidis* infection. These authors have published a work that showed the effects of *X. szentirmaii* metabolites on juvenile toads of *Bufo bufo* in 2023 and this work is an extension of that work. They have tested whether the cell-free culture media (CFCM) of bacteria affects the survival, development, and body mass of the tadpoles and checked whether the CFCM can reduce the Bd prevalence and infection intensities of tadpoles. Authors have shown that CFCM can reduce the intensity and the Bd prevalence of the infected tadpoles without compromising the survival, development, or body mass.

Bacterial secondary metabolites have been used in treating different infections caused by bacteria and fungi however using the metabolites from *X. szentirmaii* in treating chytridiomycosis is a novel and interesting approach for mitigating chytridiomycosis. There are several concerns.

We thank the Reviewer for the overall positive evaluation.

Minor concerns:

- Organizing the methods section to have a flow to make it easy for the readers to understand the approach of the work.

We made several modifications on the methods section according to the suggestions of both Reviewers. We hope that these changes made it easier for the readers to follow the flow of the work.

- Line 135 – please spell out the LBTA as this is the first time you say that.

We added the correct name (Luria Bertani Agar; line 138).

- Lines 135-137 – state indicator media used for identification of antimicrobial peptide producers and non-producers of the bacteria *X. szentirmaii* and labeled as phase I and II. It's not clear what are these labeling and did not notice these being used again in the manuscript. I suggest stating how indicator media are used to recognize antimicrobial peptide-producing bacteria.

The Reviewer is right; this labelling did not fit well to the structure and approach of this manuscript. Therefore, we removed these labels from the text. Furthermore, we added a short explanation on how the indicator media was used to recognize the antimicrobial metabolite-producing colonies (lines 141-143).

- Line 148 – incubated *X. szentirmaii* at 28C then in line 153 – have incubated at 25C.

Why have you used two different temperatures for growing the bacteria and extracting the CFCM? Please clarify and indicate if you have seen higher CFCM production at lower temperatures.

This microbe grows a bit better at 28-30 °C (Cimen et al., 2021; Lengyel et al., 2005), but 25 °C is also ideal resulting in similar CFCM capacity (Böszörményi et al., 2009). It would have been better to

grow it at 28 °C during both phases, but unfortunately, we couldn't have space capacity for that big amount in the 28 °C thermostat.

- Line 155 – I suggest stating that “antimicrobial metabolite production reached its maximum” instead of saying “stationary phase”

We agree with this modification and changed it in the text.

- Suggest selecting headings for sections appropriately. In line 131 - started methods section is “materials and methods” and again named a subsection as “experimental procedures” (in line 165). This confuses the readers.

Thank you for the Reviewer for this suggestion. We think, that the “Maintaining Bd cultures and experimental exposure” section was at the wrong place indeed, because this took place before the experiment itself. Therefore, moving it before the “Experimental procedures section” seemed to be logical, and we accomplished it (lines 169-185). However, we would keep the rest of the structure of the headings, because the heading “experimental procedures” describes the methods of the experiment specifically, but the other subheadings in the Materials and Methods section (e.g. “Culturing of bacteria”, “Maintaining Bd cultures and experimental exposure”) describe more general routine methods, performed before the experiment. If the Reviewer has a better suggestion for replacing the subheading “Experimental procedure”, we are happy to discuss it.

- There is confusion about their labeling of TGhL media. Line 206 – states TGhL and om brackets say mTGhL for the regular TGhL recipe. Authors have to carefully check their labeling of media.

We use the letter “m” before “TGhL” (or even TGhLY) as an abbreviated version of the word “medium”. Therefore, in some parts of the text, where the context requires, we spell out “medium” (and didn't use “m”), otherwise abbreviate it. This structure was consistently maintained throughout the text.

- I suggest authors carefully check the methods section again consolidate some information and reorganize the sections to improve the clarity of the approach.

We made several modifications on the methods section according to the suggestions of both Reviewers. We hope these are satisfactory and useful changes to achieve this goal of improvement. If the Reviewer feels that further modifications are needed, please point them out and we are ready to discuss them.

Major concerns:

- The major concern I have is the sample size that the authors have used for the experiment. According to the authors, they have used different numbers of tadpoles in each treatment for each site. In part 1 of the experiment, they kept 9/treatment for site 1; 8/treatment for site 2; 7/treatment for site 3; and 6/treatment for site 4. Then they continued part 2 of the experiment after 22 days of part 1 after removing five individuals from each treatment group (lines 189-191) and euthanizing them for detecting the Bd infection status of tadpoles. The number of individuals for each treatment for a site is low and if they have removed 5 tadpoles from each treatment group that brings down the number of tadpoles for testing CFCM effects really low. If they have used only 1 individual from each treatment group for detecting Bd infection status that number is still low for detecting Bd infection with higher statistical significance. This drops the rigor of their work. However, the experimental design is good and appropriate.

Apologies if we were misunderstood. Those sample sizes represented numbers of individuals from each site in one treatment, and the number after them, “in total” referred for the total sample size used from the given site. Therefore, in each treatment, all individuals per site are added up, resulting in a much higher and effective sample size than it seemed: 9 (from site 1) + 8 (from site 2) + 7 (from site 3) + 6 (from site 4) = 30 individuals in total per treatments (evenly distributed among sites). From this 30, we picked up 5 individuals randomly for assessing initial infection status per treatment.

Therefore, we had 15 unexposed individuals (assigned to Treat 1, 2 and 3) and 10 Bd-exposed individuals (assigned to Treat 4 and 5) to assess initial infection status. Consequently, we had 25 individuals per treatment to begin with the second part of the experiment (line 219). Since mortality was less, than 25 % in each treatment group, we consider the final sample sizes suitable for drawing conclusions. We reorganized the text to present this design and sample sizes more clearly (lines 203-209).

- Lines 212-214 – describe inoculating tadpoles with 750 zoospores/ml. However, the culture they used for this appeared to be a mixed culture of Bd as they have used 7d old Bd liquid culture. Authors need to be clear about how zoospores were separated (whether they separated the zoospores from the mixture before counting). The 7d old Bd mixed culture grown in culture flasks has more sporangia than spores and that may result in underestimation of the zoospore counts and different infection status.

We thank to the Reviewer for pointing this out. Since sporangia were indeed not isolated, we may have underestimated the zoospore count. However, during inoculation, we regularly shook up the Bd culture, so that we distributed the sporangia in the culture as evenly as possible among individuals. Therefore, we believe that this does not affect the reliability of our results. This missing information has been added (lines 181-182).

- Authors do not describe using an internal standard for the qPCR experiment and need one for the accuracy of the experiment which would help in accounting for the sample preparation errors. Having an internal standard is very important as they compare the treatments and sites.

We did not apply internal control for the following reasons. While internal control can be valuable indeed for detecting inhibitors and ensuring the reliability of qPCR assays, their necessity depends on factors such as the validation of extraction methods, and the presence of inhibitors in the samples. We believe, that in controlled laboratory settings with validated extraction and qPCR methods (Boyle et al., 2004) and low risk of inhibition (clean containers and filtered, reconstituted water), genomic equivalent standards as external controls should suffice. Furthermore, the inhibition of qPCR is also unlikely because of the large number of detected Bd-positive animals. Variability among sites could not be due to undetected PCR inhibition caused by the lack of internal positive control, as the individuals were collected as eggs, then continuously reared in clean, identical laboratory water, which differed only in treatments. Therefore, although there may be differences in the presence of inhibitory factors between sites, these were excluded by the laboratory approach.

- Line 271-272 – The authors claim CFCM did not affect the body mass of Bd-exposed or CFCM-treated tadpoles. However, Bd-exposed tadpoles have shown slightly reduced body mass than non-Bd-exposed tadpoles. In treatment 2,3, and the CFCM treated group, treatment 5 has slightly higher body mass than 4. So CFCM has shown a slight effect on body mass even though statistically significant. The low number of individuals used might have caused this. I suggest the authors describe the observed results and then state the statistical significance.

*The Reviewer is right, there was a slight (but statistically not significant) reduction in the body mass of individuals from Treat 4, compared to the other three treatments mentioned (Treat 2, 3 and 5). However, since Treat 4 did not contain CFCM (Table 1), this does not contradict to our conclusion that CFCM alone did not cause a reduction in body mass. Rather, it suggests that the negative effect of Bd per se on body mass in Treat 4 seems to be counteracted by the administration of CFCM, since in Treat 5, we measured similar weights as in the groups without Bd treatment (either with or without CFCM; Treat 2 and 3, Table 1). Seen in this way, CFCM application may even be beneficial, counteracting the negative impact of Bd on body mass, as we have shown in our previous study on juvenile common toads (*Bufo bufo*)(Ujszegi et al., 2023). However, due to the lack of statistical significance, we thought it is better not to address this issue in the present study. Since we previously clarified that there were still 18-21 individuals per treatment group (after mortality and assessment of initial infection status), we believe this sample size is sufficient for a robust and credible result.*

- Even though the authors describe the Bd prevalence it is not clear how did they measure the Bd prevalence in the methods section. The authors need to clarify that in the methods section. The data presented in the paper showing the effects of population on the Bd prevalence and infection intensity would be good. Suggest adding those figures.

We thank the Reviewer for this suggestion. We created a binomial variable based on whether an individual is Bd positive or not. Bd prevalence was defined as the proportion of infected individuals. Now, we have included this information in the text (lines 255-257). Also thank for the valuable comment regarding the necessity of presenting results and figures broken down by population. The significant interaction effect indeed suggests that treatment efficacy may vary across populations. According to this suggestion, we have prepared a confidence interval plot by population (see: Fig R1).

1. Figure R1: Means of Bd load with 95% Confidence intervals.

However, we would like to emphasize that the broken-down sample sizes within each population group vary considerably, especially in population 4, where only 3 non-treated and 4 CFCM-treated individuals remained. With such small sample sizes, confidence intervals widen substantially, uncertainty increases, and statistical conclusions may be biased. Therefore, we draw attention to the limited interpretability of these data, and reliable conclusions should primarily be drawn from the main effects calculated on the full sample. The figure clearly shows that in populations with larger sample sizes (e.g., population 1, where the treatment resulted in 100% recovery), confidence intervals are narrower, whereas estimates are more uncertain in small sample groups (e.g., population 4). For this reason, we have chosen not to elaborate on individual population-level analyses in the main text, as such detailed breakdowns based on very small sample sizes could be misleading and are unlikely to provide robust or generalizable conclusions.

- The observed significantly lower growth of the tadpoles housed with reconstituted soft water has not been addressed well in the discussion. Authors have tried to support this difference stating that it might be due to the nutrient media in the treatments. Even though microbial media can have some effects providing some nutrients to tadpoles, it is difficult to accept that this is the major reason for the very prominent growth reduction observed in tadpoles reared in RSW. The method used to make RSW might have affected the growth of the tadpoles and authors need to address this.

The RSW was prepared according to the following recipe: 48 mg NaHCO₃, 30 mg CaSO₄ × 2 H₂O, 61 mg MgSO₄ × 7 H₂O, 2 mg KCl added to 1 L reverse-osmosis filtered, UVsterilized tap water (USEPA, 2002). This has now been added directly to the manuscript for clarity. It is unlikely that the difference among the groups is due to the preparation of the RSW, because all individuals were reared in the same RSW, except that in the other treatment groups, sterile (Treat 2 and 3) or Bd-containing (Treat 4 and 5) medium was added too. We consider the benign effect of the nutrients from the media to be the most likely direct explanation, providing them with additional food source for growth. It is

also possible that osmolality changed due to the addition of medium or the medium exerted a buffering effect, counteracting pH fluctuations due to the CO₂ accumulation over time. Therefore, the addition of media probably indirectly influenced the growth of the individuals too, but since we did not measure these variables, this remains only a rough speculation. However, now we have included these alternative explanations in the discussion (lines 310-313).

- The discussion section needs the author's attention and suggests revising it. The conclusion of the study is not well articulated and not clear as authors make different arguments that contradict each other.

We made an effort to be more comprehensible and conclusive with incorporating several modifications based on the comments and suggestions of both Reviewers. We hope this effort was satisfactory and useful. If the Reviewer feels that there is further work to be done on this section, please do not hesitate to indicate which discrepancies are concerned.

- The authors have stated in the abstract that the reduction of the intensity and Bd prevalence by CFCM was noticed only in the case of one population out of the four populations that they used. However, they have not shown this data or discussed this in the discussion. They have not stated which population was affected by the CFCM. It will be interesting to understand this.

The Reviewer is right; this information was missing from the manuscript. Now we have added in the discussion (lines 324-329).

- I suggest authors test the effects using just one population as that would remove the population variability.

Thank you for your suggestion to test the effects using just one population in order to remove population variability. Our results showed that the treatment influenced only individuals from a single population, while others were not significantly affected. In our statistical model, population was a significant factor. We anticipated such differences among populations in advance, which is precisely why we included population as a factor in our model. Excluding population variability would not only overlook biologically meaningful variation but also run counter to our aim of understanding how treatment effects may differ among populations. Therefore, we believe it is important to retain all populations in the analysis and to model population as a factor, rather than omitting it, or restricting the analysis to a single population. Furthermore, restricting analysis to the only population where the treatment worked would make no sense, because the effectiveness was 100% (line 327), leaving no residual variance to analyse statistically (Fig. R2).

Figure R2: In the population (Site 1, Haluskási-út), where CFCM treatment worked, all individuals cleared the infection, leaving no variance to statistically analyse.

Literature cited

- Böszörményi, E., Érsek, T., Fodor, A., Fodor, A. M., Földes, L. S., Hevesi, M., Hogan, J. S., Katona, Z., Klein, M. G., Kormány, A., Pekár, S., Szentirmai, A., Sztaricskai, F., & Taylor, R. A. J. (2009). Isolation and activity of *Xenorhabdus* antimicrobial compounds against the plant pathogens *Erwinia amylovora* and *Phytophthora nicotianae*. *Journal of Applied Microbiology*, *107*(3), 746–759. <https://doi.org/10.1111/j.1365-2672.2009.04249.x>
- Boyle, D. G., Boyle, D. B., Olsen, V., Morgan, J. A. T., & Hyatt, A. D. (2004). Rapid quantitative detection of chytridiomycosis (*Batrachochytrium dendrobatidis*) in amphibian samples using real-time Taqman PCR assay. *Diseases of Aquatic Organisms*, *60*(2), 141–148. <https://doi.org/10.3354/dao060141>
- Cimen, H., Touray, M., Gulsen, S. H., Erincik, O., Wenski, S. L., Bode, H. B., Shapiro-Ilan, D., & Hazir, S. (2021). Antifungal activity of different *Xenorhabdus* and *Photorhabdus* species against various fungal phytopathogens and identification of the antifungal compounds from *X. szentirmaii*. *Applied Microbiology and Biotechnology*, *105*(13), 5517–5528. <https://doi.org/10.1007/s00253-021-11435-3>
- Lengyel, K., Lang, E., Fodor, A., Szállás, E., Schumann, P., & Stackebrandt, E. (2005). Description of four novel species of *Xenorhabdus*, family *Enterobacteriaceae*: *Xenorhabdus budapestensis* sp. nov., *Xenorhabdus ehlersii* sp. nov., *Xenorhabdus innexi* sp. nov., and *Xenorhabdus szentirmaii* sp. nov. *Systematic and Applied Microbiology*, *28*(2), 115–122. <https://doi.org/10.1016/j.syapm.2004.10.004>
- Ujszegi, J., Boros, Z., Fodor, A., Vajna, B., & Hettyey, A. (2023). Metabolites of *Xenorhabdus* bacteria are potent candidates for mitigating amphibian chytridiomycosis. *AMB Express*, *13*(1), 88. <https://doi.org/10.1186/s13568-023-01585-0>
- USEPA. (2002). Section 7: Dilution water. In *Methods for Measuring the Acute Toxicity of Effluents and Receiving Waters to Freshwater and Marine Organisms* (5th ed., Issue October, p. 33). Office of Water, U.S. Environmental Protection Agency, Washington, D.C., EPA-821-R-02-012.
- Zacho, C. M. (2023). *Whole exome analyses of population structure and genotype-phenotype association in the Bombina hybrid zone*. University of Copenhagen.

Re: Spectrum00826-25R1 (Mitigation of chytridiomycosis on *Bombina variegata* tadpoles with the antimicrobial metabolites of the bacterium *Xenorhabdus szentirmai*)

Dear Dr. János Ujszegi:

Your manuscript has been accepted, and I am forwarding it to the ASM production staff for publication. Your paper will first be checked to make sure all elements meet the technical requirements. ASM staff will contact you if anything needs to be revised before copyediting and production can begin. Otherwise, you will be notified when your proofs are ready to be viewed.

Editor's comment:

Based on the received reviews, I would like to request the following minor changes during the proofreading. Please briefly mention in one sentence in the discussion and emphasizing the need for a larger sample size.

Sincerely,
Renato Kovacs
Editor
Microbiology Spectrum